# Sensorless Fault-Tolerant Control of Dual Three-Phase Permanent Magnet Synchronous Motor

**Fan Cao [1,*], Haifeng Lu [1], Yonggang Meng [2] and Dawei Gao [3]**

1. Department of Electrical Engineering, Tsinghua University, Beijing 100084, China; luhaifeng@mail.tsinghua.edu.cn
2. Lianchuang Automotive Electronics, Shanghai 200082, China; mengyonggang@dias.com.cn
3. School of Vehicle and Mobility, Tsinghua University, Beijing 100084, China; dwgao@mail.tsinghua.edu.cn
* Correspondence: caof19@mails.tsinghua.edu.cn

**Abstract:** Dual three-phase permanent magnet synchronous motors (DTPMSM) are used in the steer-by-wire system of electric vehicles that require high reliability. Multiple faults should be considered for the steering system, such as open-circuit faults and speed sensor faults. However, the current speed sensorless control methods of the dual three-phase motor are mainly derived from the promotion of the three-phase motor. They fail when an open-circuit fault occurs, leading to the failure of fault-tolerant control. Researchers have noticed this problem and proposed many methods, but they are very complicated and computationally intensive. This paper proposes one type of improved model reference adaptive system (MRAS). By adding certain fault-related restraints to the output of the adjustable model, speed sensorless control can automatically fit the open-circuit fault and estimate accurately even if an open-circuit fault occurs, which makes sure the whole system continues to operate. Simulation results are presented that contain normal operation, open-circuit fault operation, fault-tolerant control operation, and the whole process from start to fault-tolerant operation. The results show that no matter what period the motor is in, the improved speed sensor can accurately estimate the motor speed and position. The improved model reference adaptive system is significant for improving the reliability of the motor steering system and ensuring the safety of people and property.

**Keywords:** connected EV; modeling; open circuit; permanent magnet motor; power steering

## 1. Introduction

Because of high power density, high reliability, and low torque ripple, dual three-phase motors are widely used in electric vehicles, ship driving, and other fields. When open-circuit faults occur, even if the neural point is not connected to the midpoint of the bus, it still can generally operate through proper fault-tolerant control strategies. The motor winding arrangement is shown in Figure 1, and the structure diagram of using two three-phase inverters to drive dual three-phase motors is shown in Figure 2.

It has been many years since the emergence of multi-phase motors and the proposal of the related fault-tolerant control strategies. The methods of fault-tolerant control have been relatively mature. In the 1990s, Y.F Zhao and Thomas A. Lipo proposed a vector space decomposition (VSD) method and adopted four-vector SVPWM modulation. In 1996, they proposed a fault-tolerant control strategy that established the mathematical model of the multi-phase motor under fault conditions. Then, they performed a secondary coordinate transformation to eliminate coupling between the $d$ axis and $q$ axis [1].

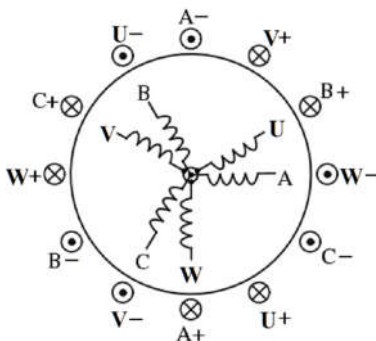

**Figure 1.** Block diagram of the fault-tolerant control strategy.

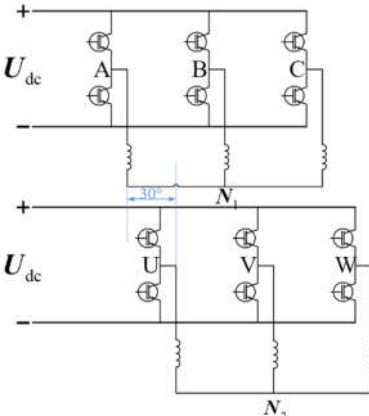

**Figure 2.** Dual three-phase motor system powered by dual voltage source inverters.

After the 2000s, more and more scholars devoted themselves to research related to fault-tolerant control. They combined novel control strategies with original motor control to eliminate the impact of the open-circuit fault, such as model reference control, deadbeat control, and slide-mode control. In the past two years, scholars have improved these methods. Hugo Guzman used model-based predictive current control to replace PI control to achieve a similar control effect during regular operation [2]. Tao used a finite control set model predictive control for a faulty five-phase PMSM, drastically reducing the calculation time and improving steady-state performance [3]. Yixiao Luo proposed a novel deadbeat current control-based model predictive control. It can not only improve dynamic performance and steady-state accuracy but also reduce the amount of calculation [4]. Lei Xu used a deadbeat controller to replace the traditional PI controller in the closed-loop control of a faulty six-phase motor, but compared with the increased calculation, the performance improvement is not apparent enough [5]. After 30 to 40 years of development, fault-tolerant control theories are relatively mature. The fault-tolerant control strategies can be divided into three main categories: decoupling control, proposed by Y.F Zhao; current optimization control to maintain circular flux; and novel robust control. This paper uses the original decoupling control as the fault-tolerant strategy.

In the 2010s, scholars noticed that other fundamental control methods would fail because of the open-circuit fault. They began to consider improving to make them work even if they fail, such as MTPA, speed sensorless control, and field weakening control. Take speed sensorless control as an example. It requires the voltage and current signals of the motor. Without open-circuit faults, these conventional speed sensorless control methods perform well. When an open-circuit fault occurs, the structure of the motor changes and the observation accuracy is inevitably affected. In the past two years, there have been few research results concerning this problem. Alberto Gaeta analyzed the three-phase motor

under one phase open. Then, he compared three types of speed sensorless control strategies considering the open-circuit fault: model-based sensorless estimation I, model-based sensorless estimation II, and carrier-signal-injection-based sensorless technique. The results show that the third method has the highest accuracy, and it is necessary to improve the conventional method to fit the fault [6]. Zhong Peng and Zicheng Liu combined the virtual winding method and full-order observer to realize the observation of speed and angle under normal and fault conditions, which is effectively proved by simulation and experiment [7]. Yonggang Li improved the sliding mode velocity observer under open phase fault. It is effective, but the chattering problem still exists [8]. Jinquan Xu proposed a new sensorless control based on the robust observer, nonorthogonal phase-locked loop (PLL), and variable phase delay compensation. They performed the method on a faulty six-phase PMSM, and it could guarantee excellent speed control performance even under the postfault condition [9]. Although researchers have noticed that sensorless control methods fail when an open-circuit fault occurs on multi-phase motor and proposed serval solutions, they are complicated and computationally intensive.

In order to realize the identification under postfault conditions, this paper proposes an improved model reference adaptive system. This method does not need to change the adjustable model in MRAS according to different faults; rather, adding some fault-related restrictions to the output of the adjustable model realizes the identification under normal and fault conditions. The fault-tolerant method establishes the mathematical model and performing vector control. A simulation model with MATLAB/Simulink is established, and the proposed method is simulated to verify the effectiveness of the algorithm on different occasions.

## 2. Fault-Tolerant Control Strategy

### 2.1. Establishment of Faulty DTPMSM

As shown in Figure 2, the neutral points of two windings in the dual three-phase motor are not connected nor to the midpoint of the DC bus. Although the harmonic current of this connection in regular operation is slightly bigger than when the neutral point is connected, it can avoid zero-sequence currents between the two sets of windings under open-circuit fault [10].

Taking W-phase open circuit as an example, this paper establishes the mathematical model of the faulty motor and carries on vector control. For the W-phase open-circuit DTPMSM system, the voltage equation and flux equation are as follows:

$$U_s = R_s I_s + p \psi_s \,. \tag{1}$$

$$\psi_s = L_s(\theta) I_s + \Gamma(\theta) \psi_m \,. \tag{2}$$

where $U_s = [u_A \ u_B \ u_C \ u_U \ u_V]^T$, stator voltage matrix; $R_s = \mathrm{diag}[R_s \ R_s \ R_s \ R_s \ R_s]^T$, stator resistance matrix; $I_s = [i_A \ i_B \ i_C \ i_U \ i_V]^T$, stator current matrix; p, differential operator; $\psi_s = [\psi_A \ \psi_B \ \psi_C \ \psi_U \ \psi_V]^T$, stator flux matrix; $\Gamma(\theta) = [\cos(\theta) \ \cos(\theta - \frac{2\pi}{3}) \ \cos(\theta + \frac{2\pi}{3})$ $\cos(\theta - \frac{\pi}{6}) \ \cos(\theta - \frac{5\pi}{6})]^T$; $\psi_m$, permanent magnet flux; $L_s(\theta)$, stator inductance matrix.

$$
\begin{aligned}
\boldsymbol{L}_s(\boldsymbol{\theta}) = \boldsymbol{L}_{aal}\boldsymbol{I}_5 + \boldsymbol{L}_0
\begin{bmatrix}
1 & \cos(\frac{2\pi}{3}) & \cos(\frac{4\pi}{3}) & \cos(\frac{\pi}{6}) & \cos(\frac{5\pi}{6}) \\
\cos(\frac{4\pi}{3}) & 1 & \cos(\frac{2\pi}{3}) & \cos(\frac{3\pi}{2}) & \cos(\frac{\pi}{6}) \\
\cos(\frac{2\pi}{3}) & \cos(\frac{4\pi}{3}) & 1 & \cos(\frac{5\pi}{6}) & \cos(\frac{3\pi}{2}) \\
\cos(\frac{\pi}{6}) & \cos(\frac{3\pi}{2}) & \cos(\frac{5\pi}{6}) & 1 & \cos(\frac{2\pi}{3}) \\
\cos(\frac{5\pi}{6}) & \cos(\frac{\pi}{6}) & \cos(\frac{3\pi}{2}) & \cos(\frac{4\pi}{3}) & 1
\end{bmatrix} \\[2mm]
+ \boldsymbol{L}_2
\begin{bmatrix}
\cos(2\boldsymbol{\theta}) & \cos 2(\boldsymbol{\theta}-\frac{\pi}{3}) & \cos 2(\boldsymbol{\theta}+\frac{\pi}{3}) & \cos 2(\boldsymbol{\theta}-\frac{\pi}{12}) & \cos 2(\boldsymbol{\theta}-\frac{5\pi}{12}) \\
\cos 2(\boldsymbol{\theta}-\frac{\pi}{3}) & \cos 2(\boldsymbol{\theta}+\frac{\pi}{3}) & \cos(2\boldsymbol{\theta}) & \cos 2(\boldsymbol{\theta}-\frac{5\pi}{12}) & \cos 2(\boldsymbol{\theta}+\frac{\pi}{4}) \\
\cos 2(\boldsymbol{\theta}+\frac{\pi}{3}) & \cos(2\boldsymbol{\theta}) & \cos 2(\boldsymbol{\theta}-\frac{\pi}{3}) & \cos 2(\boldsymbol{\theta}+\frac{\pi}{4}) & \cos 2(\boldsymbol{\theta}-\frac{\pi}{12}) \\
\cos 2(\boldsymbol{\theta}-\frac{\pi}{12}) & \cos 2(\boldsymbol{\theta}-\frac{5\pi}{12}) & \cos 2(\boldsymbol{\theta}+\frac{\pi}{4}) & \cos 2(\boldsymbol{\theta}-\frac{\pi}{6}) & \cos 2(\boldsymbol{\theta}-\frac{\pi}{2}) \\
\cos 2(\boldsymbol{\theta}-\frac{5\pi}{12}) & \cos 2(\boldsymbol{\theta}+\frac{\pi}{4}) & \cos 2(\boldsymbol{\theta}-\frac{\pi}{12}) & \cos 2(\boldsymbol{\theta}-\frac{\pi}{2}) & \cos 2(\boldsymbol{\theta}+\frac{\pi}{6})
\end{bmatrix}
\end{aligned}
\tag{3}
$$

where $\boldsymbol{L}_{aal}$ is the leakage inductance of the stator winding, $\boldsymbol{L}_0$ is the average value of the main self-inductance, and $\boldsymbol{L}_2$ is the second harmonic amplitude of the main self-inductance.

The coordinate transformation matrix from the five-phase stationary coordinate system to the $\boldsymbol{\alpha}-\boldsymbol{\beta}-\boldsymbol{z1}-\boldsymbol{z2}-\boldsymbol{z3}$ coordinate system is [10]:

$$
\boldsymbol{T}_{5s} = \frac{1}{3}
\begin{bmatrix}
1 & -\frac{1}{2} & -\frac{1}{2} & \frac{\sqrt{3}}{2} & -\frac{\sqrt{3}}{2} \\
0 & \frac{\sqrt{3}}{2} & -\frac{\sqrt{3}}{2} & 0 & 0 \\
1 & -\frac{1}{2} & -\frac{1}{2} & -\frac{\sqrt{3}}{2} & \frac{\sqrt{3}}{2} \\
1 & 1 & 1 & 0 & 0 \\
0 & 0 & 0 & 1 & 1
\end{bmatrix}
\tag{4}
$$

The $\boldsymbol{\alpha}-\boldsymbol{\beta}$ sub-plane involved in the electromechanical energy conversion needs to be transformed to realize the transformation from a stationary coordinate system to a rotating coordinate system. The coordinate transformation matrix is as follows:

$$
\boldsymbol{P}_5 =
\begin{bmatrix}
\cos\theta & \sin\theta & 0 \\
-\sin\theta & \cos\theta & 0 \\
0 & 0 & I_3
\end{bmatrix}
\tag{5}
$$

The final transformation matrix from the five-phase stationary coordinate system to the two-phase rotating coordinate system is:

$$
\boldsymbol{T}_5 = \boldsymbol{P}_5 \boldsymbol{T}_{5s}
\tag{6}
$$

After the coordinate transformation matrix is calculated, the modeling process of the faulty motor is similar to that of the normal one, which uses $\boldsymbol{T}_5$ to multiply the left and right sides of the voltage Equation (1) and the flux Equation (2).

$$T_5 U_s = T_5 R_s I_s + T_5 \frac{d\psi_s}{dt}$$

$$= (T_5 R_s T_5^{-1})(T_5 I_s) + \frac{d(T_5 \psi_s)}{dt} - \left(\frac{dT_5}{dt} T_5^{-1}\right)(T_5 \psi_s) \tag{7}$$

$$= R_{dq} I_{dq} + \frac{d\psi_{dq}}{dt} - \Omega \psi_{dq}$$

$$T_5 \psi_s = T_5 L_s I_s + T_5 \psi_m$$
$$= (T_5 L_s T_5^{-1})(T_5 I_s) + T_5 \psi_m \tag{8}$$
$$= L_{dq} I_{dq} + \psi_{dqm}$$

If only considering the $\mathbf{d-q}$ sub-plane where energy conversion occurs, then $U_{dq} = T_5 U_s = [u_d \quad u_q]^T$ ; $R_{dq} = T_5 R_s T_5^{-1} = R_s I_2$ ; $I_{dq} = T_5 I_s = [i_d \quad i_q]^T$ ; $\psi_{dq} = T_5 \psi_s = [\psi_d \quad \psi_q]^T$.

$\Omega$ is the velocity matrix:

$$\Omega = \frac{dT_5}{dt} T_5^{-1} = \omega \begin{bmatrix} 0 & 1 \\ -1 & 0 \end{bmatrix} \tag{9}$$

The inductance matrix $L_{dq}$ is:

$$L_{dq} = T_5 L_s T_5^{-1} = L_{aal} I_2 + 3 \begin{bmatrix} L_{aad} & 0 \\ 0 & L_{aaq} \end{bmatrix} A(\theta) \tag{10}$$

where $L_{aal}$ is leakage inductance of stator winding, and $L_{aad}$ and $L_{aaq}$ are the main inductance of d–q axes.

$$A(\theta) = \begin{bmatrix} 0.75 + 0.25\cos 2\theta & -0.25\sin 2\theta \\ -0.25\sin 2\theta & 0.75 - 0.25\cos 2\theta \end{bmatrix}$$

The flux is:

$$\psi_{dqm} = T_5 \psi_m = \psi_{fd} \begin{bmatrix} 0.75 + 0.25\cos 2\theta \\ -0.25\sin 2\theta \end{bmatrix} \tag{11}$$

Finally, the stator voltage equation of the faulty motor in the $d–q$ frame can be described as:

$$\begin{bmatrix} u_d \\ u_q \end{bmatrix} = \begin{bmatrix} R_s & 0 \\ 0 & R_s \end{bmatrix} \begin{bmatrix} i_d \\ i_q \end{bmatrix} + \left( \begin{bmatrix} L_{aal} & 0 \\ 0 & L_{aal} \end{bmatrix} + A(\theta) \begin{bmatrix} 3L_{aad} & 0 \\ 0 & 3L_{aaq} \end{bmatrix} \right) \frac{d}{dt} \begin{bmatrix} i_d \\ i_q \end{bmatrix}$$
$$+ \omega \left( A(\theta) \begin{bmatrix} 0 & -3L_{aaq} \\ 3L_{aad} & 0 \end{bmatrix} + \begin{bmatrix} 0 & -L_{aal} \\ L_{aal} & 0 \end{bmatrix} \right) \begin{bmatrix} i_d \\ i_q \end{bmatrix} + \begin{bmatrix} -0.25\sin 2\theta \\ 0.75 - 0.25\cos 2\theta \end{bmatrix} \omega \psi_{fd} \tag{12}$$

The voltage equation of the $z1$-axis is:

$$u_{z1} = R_s i_{z1} + L_{aal} \frac{di_{z1}}{dt} \tag{13}$$

The $z2$-axis and $z3$-axis are zero-sequence current components due to the isolated neutrals. Therefore, it is not necessary to perform closed-loop control, just to set the voltage to zero.

*2.2. Vector Control of Faulty DTPMSM*

According to the mathematical model of the faulty motor above, the inductance matrix and flux matrix contain terms related to the double frequency of the speed, which is constantly changing with time. Furthermore, there is a coupling between the *d-q* axes. In order to eliminate the influence of these factors, multiply both sides of the voltage matrix by $A(\theta)^{-1}$ to perform the secondary transformation, and the result is as follows:

$$
A(\theta)^{-1}\begin{bmatrix} u_d \\ u_q \end{bmatrix} = \begin{bmatrix} 1.5R_s & 0 \\ 0 & 1.5R_s \end{bmatrix}\begin{bmatrix} i_d \\ i_q \end{bmatrix} + \begin{bmatrix} 1.5L_{aal}+3L_{aad} & 0 \\ 0 & 1.5L_{aal}+3L_{aaq} \end{bmatrix}\frac{d}{dt}\begin{bmatrix} i_d \\ i_q \end{bmatrix}
$$
$$
+ \omega\begin{bmatrix} 0 & -(1.5L_{aal}+3L_{aaq}) \\ 1.5L_{aal}+3L_{aad} & 0 \end{bmatrix}\begin{bmatrix} i_d \\ i_q \end{bmatrix} + \begin{bmatrix} 0 \\ 1 \end{bmatrix}\omega\psi_{fd} + F(\theta)\begin{bmatrix} i_d \\ i_q \end{bmatrix}. \quad (14)
$$

where:

$$
F(\theta) = \frac{R_s}{2}\begin{bmatrix} -\cos 2\theta & \sin 2\theta \\ \sin 2\theta & \cos 2\theta \end{bmatrix} + \frac{L_{aal}}{2}\begin{bmatrix} -\cos 2\theta & \sin 2\theta \\ \sin 2\theta & \cos 2\theta \end{bmatrix}\frac{d}{dt} + \frac{\omega L_{aal}}{2}\begin{bmatrix} \sin 2\theta & \cos 2\theta \\ \cos 2\theta & -\sin 2\theta \end{bmatrix}.
$$

Define:

$$
\begin{bmatrix} u_{d1} \\ u_{q1} \end{bmatrix} = A(\theta)^{-1}\begin{bmatrix} u_d \\ u_q \end{bmatrix} = \frac{1}{2}\begin{bmatrix} 3-\cos 2\theta & \sin 2\theta \\ \sin 2\theta & 3+\cos 2\theta \end{bmatrix}\begin{bmatrix} u_d \\ u_q \end{bmatrix}. \quad (15)
$$

$F(\theta)$ is the component related to the stator resistance, leakage inductance, and double frequency of the speed. If $F(\theta)=0$, then $u_{d1}$ and $u_{q1}$ can be completely decoupled. In this regard, the feedforward decoupling compensation method can introduce a compensation term in the current loop and finally realize the vector control of the W-phase open-circuit faulty motor.

The block diagram of the vector control of DTPMSM with one-phase open-circuit fault for secondary coordinate transformation is shown in Figure 3.

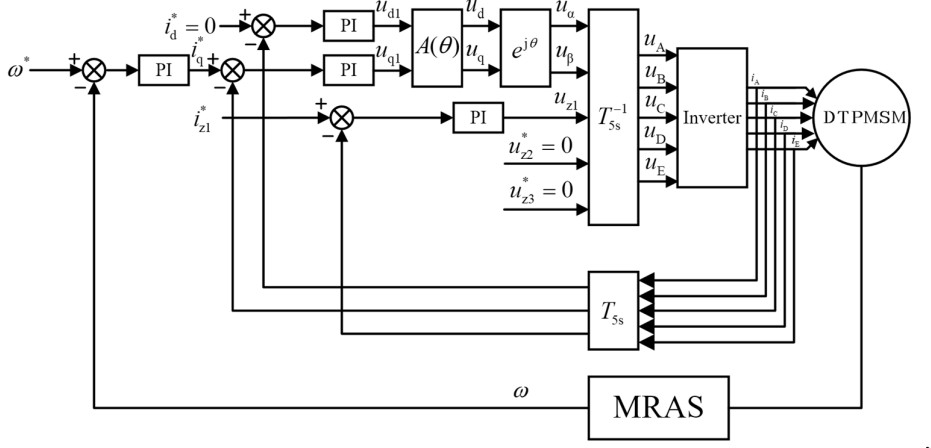

**Figure 3.** Block diagram of the fault-tolerant control strategy.

## 3. Proposed Speed Estimating Method

When in symmetrical operation, MRAS can accurately estimate the speed and position. Once one phase becomes open, the motor changes from a symmetrical system to an asymmetrical one, and conventional MRAS fails. The strategy proposed in this paper is to maintain the adjustable model in MRAS as a dual three-phase motor without any fault and add some fault-related constraints to the output of the model, which makes it automatically adapt to the motor structure and estimate correctly.



### 3.1. Original MRAS

MRAS consists of three parts: reference model, adjustable model, and adaptive law. The reference model represents the actual operating state of the system and does not contain any unknown parameters. The adjustable model has the same physical meaning as the reference model, but it contains parameters to be estimated. The two models work at the same time. The difference between their outputs is to adjust the parameters in the adjustable model according to the appropriate adaptive law. The output of the adjustable model tracks the output of the reference one. The adaptive formula is calculated according to the stability principle. The asymptotic convergence of the system is guaranteed by Popov or Lyapunov superstability [11].

The dynamic voltage equation of the DTPMSM in the d–q coordinate system is:

$$\begin{bmatrix} u_d \\ u_q \end{bmatrix} = R_s \begin{bmatrix} i_d \\ i_q \end{bmatrix} + \begin{bmatrix} L_D & 0 \\ 0 & L_Q \end{bmatrix} \frac{d}{dt} \begin{bmatrix} i_d \\ i_q \end{bmatrix} + \omega \begin{bmatrix} 0 & -L_Q \\ L_D & 0 \end{bmatrix} \begin{bmatrix} i_d \\ i_q \end{bmatrix} + \omega \begin{bmatrix} 0 \\ \psi_{fd} \end{bmatrix} \tag{16}$$

Rewrite the above equation as the state equation and choose stator current as the state variable:

$$\frac{d}{dt} \begin{bmatrix} i_d + \dfrac{\psi_{fd}}{L_D} \\ i_q \end{bmatrix} = \begin{bmatrix} -\dfrac{R_s}{L_D} & \omega \dfrac{L_Q}{L_D} \\ -\omega \dfrac{L_D}{L_Q} & -\dfrac{R_s}{L_Q} \end{bmatrix} \begin{bmatrix} i_d + \dfrac{\psi_{fd}}{L_D} \\ i_q \end{bmatrix} + \frac{1}{L_D L_Q} \begin{bmatrix} L_D u_d + \dfrac{L_Q}{L_D} R_s \psi_{fd} \\ L_Q u_q \end{bmatrix} \tag{17}$$

The adjustable model is written as follows:

$$\frac{d}{dt} \begin{bmatrix} \hat{i}_d^* \\ \hat{i}_q^* \end{bmatrix} = \begin{bmatrix} -\dfrac{R_s}{L_D} & \hat{\omega} \dfrac{L_Q}{L_D} \\ -\hat{\omega} \dfrac{L_D}{L_Q} & -\dfrac{R_s}{L_Q} \end{bmatrix} \begin{bmatrix} \hat{i}_d^* \\ \hat{i}_q^* \end{bmatrix} + \frac{1}{L_D L_Q} \begin{bmatrix} u_d^* \\ u_q^* \end{bmatrix} \tag{18}$$

where $i_d^* = i_d + \dfrac{\psi_{fd}}{L_D}$, $i_q^* = i_q$, $u_d^* = L_D u_d + \dfrac{L_Q}{L_D} R_s \psi_{fd}$, $u_q^* = L_Q u_q$. According to Popov's superstability theory, the adaptive law can be expressed as:

$$\hat{\omega} = k_p (i_d^* \hat{i}_q^* - i_q^* \hat{i}_d^*) + \int_0^t k_i (i_d^* \hat{i}_q^* - i_q^* \hat{i}_d^*) d\tau + \hat{\omega}(0) \tag{19}$$

Rewrite as PI regulator form:

$$\hat{\omega} = (k_p + \frac{k_i}{s})(i_d^* \hat{i}_q^* - i_q^* \hat{i}_d^*) \tag{20}$$

The estimated value of the rotor electrical angle can be calculated by integrating the speed as follows:

$$\hat{\theta} = \int_0^t \hat{\omega} dt + \hat{\theta}(0) \tag{21}$$

### 3.2. Proposed Improved MRAS

As mentioned above, MRAS can accurately identify the speed and angle of the motor when there is no fault, or only the speed sensor fails, and it has high steady-state accuracy and suitable dynamic performance. However, once an open-circuit fault occurs, the motor structure changes and the performance of MRAS is affected. It is thus necessary to improve MRAS to adapt to other faults and expand application scenarios of speed sensorless control.

The method adopted in this paper is to keep the adjustable model in MRAS as trouble-free DTPMSM unchanged and add some fault-related constraints to the output of the adjustable model to realize the identification of the speed of the faulty motor.

The inverse Clarke transformation is:

$$
\begin{bmatrix} i_A \\ i_B \\ i_C \\ i_U \\ i_V \\ i_W \end{bmatrix} = \frac{1}{3}
\begin{bmatrix}
1 & 0 & 1 & 0 & 1 & 0 \\
-\frac{1}{2} & \frac{\sqrt{3}}{2} & -\frac{1}{2} & -\frac{\sqrt{3}}{2} & 1 & 0 \\
-\frac{1}{2} & -\frac{\sqrt{3}}{2} & -\frac{1}{2} & \frac{\sqrt{3}}{2} & 1 & 0 \\
\frac{\sqrt{3}}{2} & \frac{1}{2} & -\frac{\sqrt{3}}{2} & \frac{1}{2} & 0 & 1 \\
-\frac{\sqrt{3}}{2} & \frac{1}{2} & \frac{\sqrt{3}}{2} & \frac{1}{2} & 0 & 1 \\
0 & -1 & 0 & -1 & 0 & 1
\end{bmatrix}
\begin{bmatrix} i_\alpha \\ i_\beta \\ i_{z1} \\ i_{z2} \\ i_{o1} \\ i_{o2} \end{bmatrix}
\tag{22}
$$

Taking W-phase open-circuit fault as an example, the W-phase current is 0; that is:

$$
i_W = -i_\beta - i_{z2} + i_{o2} = 0
\tag{23}
$$

Keeping the zero-sequence current $i_{o2} = 0$, only one constraint condition $i_{z2} = -i_\beta$ needs to be added to the output of the adjustable model to realize the adaptation to the W-phase open fault. For other open-phase open, a similar method is applicable so that MRAS can accurately identify the speed and position under fault conditions.

The control block diagram of improved MRAS is shown in Figure 4.

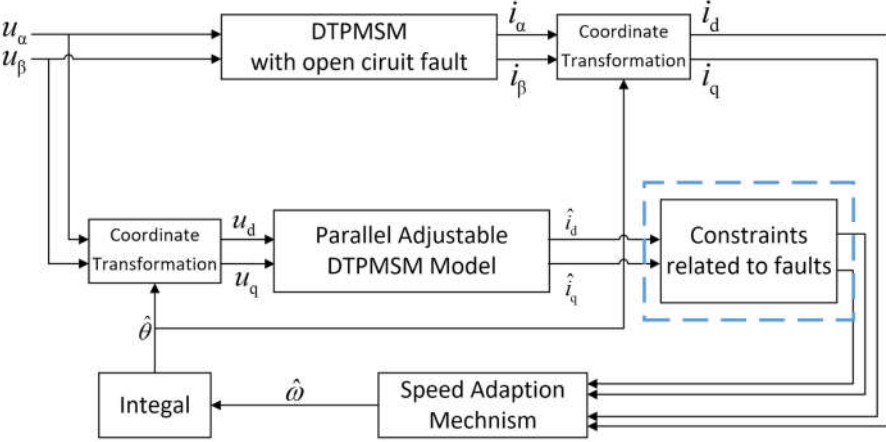

**Figure 4.** Block diagram of the fault-tolerant control strategy.

## 4. Results and Discussion

In order to verify the effectiveness of the fault-tolerant control strategy and the proposed improved MRAS speed sensorless control strategy, we conducted a simulation analysis on a built-in DTPMSM based on the Simulink toolbox of MATLAB. The parameters of the motor are shown in Table 1. The switching frequency of the inverter is 20 kHz. Load torque is 150 $\mathrm{N \cdot m}$. The solver type is a fixed step ode45 algorithm. The sample time is $1 \times 10^{-6}\,s$.

**Table 1.** Parameters of the DTPMSM in the experiment.

| Parameter | Value | Parameter | Value |
|---|---|---|---|
| Pole Number | 4 | Rated Voltage | 540 V |
| Stator Resistance $R$ | 0.0274 Ω | Rated Current | 180 A |
| Stator Leakage Inductance $L_{aal}$ | 0.06 mH | Maximum Current | 360 A |
| $d$-axis Main Inductance $L_{aad}$ | 0.242367 mH | Rated Speed | 3000 rpm |
| $q$-axis Main Inductance $L_{aaq}$ | 0.490018 mH | Rated Torque | 255 N·m |
| Permanent Magnet Flux $\psi_{fd}$ | 0.095 Wb | | |

The simulation results with no faults are shown in Figure 5. The speed is set to 3000 r/min. Under regular operation, the speed observed by MRAS is almost the same as the motor's actual speed. The current and the torque are stable.

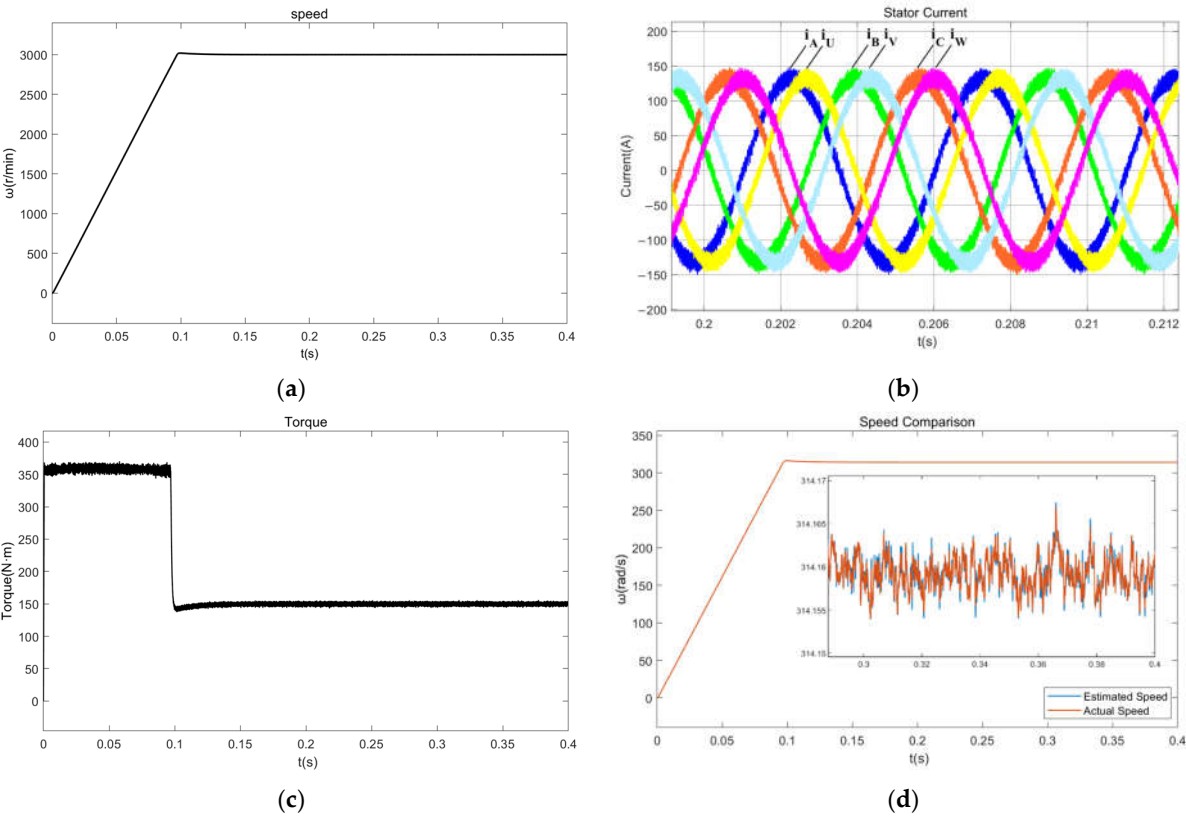

**Figure 5.** Simulation results in normal mode. (**a**) Speed; (**b**) phase current; (**c**) torque; (**d**) speed comparison.

Figures 6 and 7 show the simulation results after open-circuit fault and after the fault-tolerant operation. From Figure 6, under one-phase open-circuit fault, the torque and the speed have an apparent secondary pulsation, and the motor flux is almost elliptical. The current of each phase is not a sine wave, and their amplitudes and phases have changed.

After the fault-tolerant control strategy is performed, as shown in Figure 7, the pulsation fades away, making the torque stable. The current of each phase changes back to the sine wave, and the motor flux is a perfect circle. The whole system becomes stable.

Whether open-circuit fault occurs or not, the improved MRAS can accurately estimate the speed and angle of the motor. If there is no improved speed estimation method, the motor will lose control and influence the security of the whole electric vehicle. After

the method is performed, the speed identification operates normally, which ensures the regular operation of sensorless control on multiple occasions.

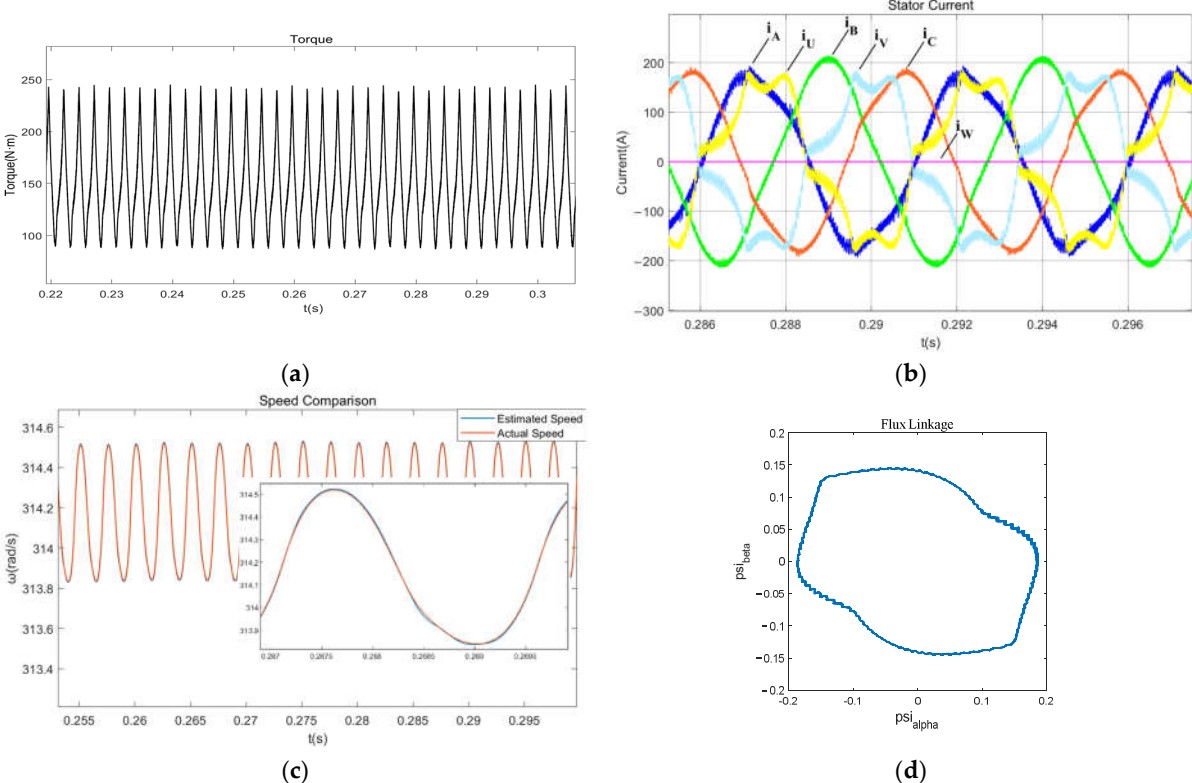

**Figure 6.** Simulation results under W-phase open circuit. (**a**) Torque; (**b**) phase current; (**c**) speed comparison; (**d**) flux.

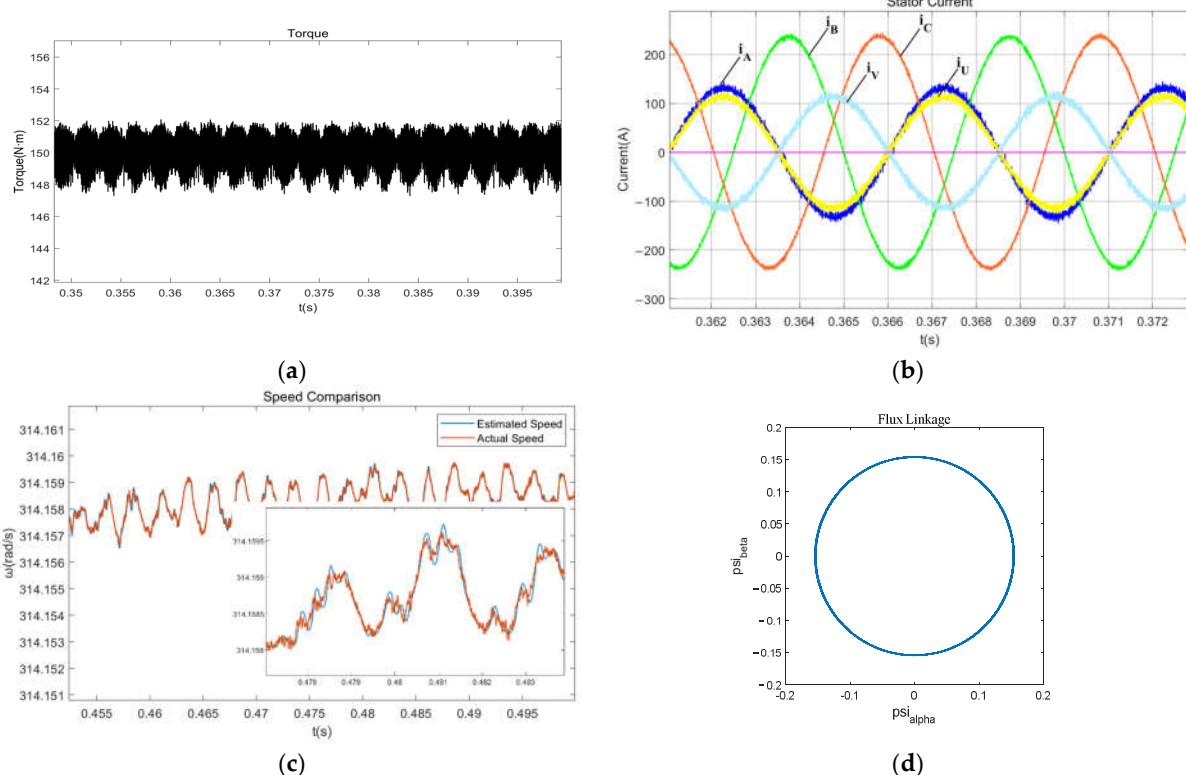

**Figure 7.** Simulation results under fault-tolerant control. (**a**) Torque; (**b**) phase current; (**c**) speed comparison; (**d**) flux.

Figure 8 shows the whole process from start-up to fault, and finally to fault-tolerant operation under improved MRAS speed sensorless control. The fault-tolerant control strategy ensures the stability of the whole system. Even if the load is increased or reduced, the system can rapidly adapt to the difference. No matter in which period, the proposed speed estimation method can accurately calculate the speed and angle of the motor, which expands the application scenarios of speed sensorless control.

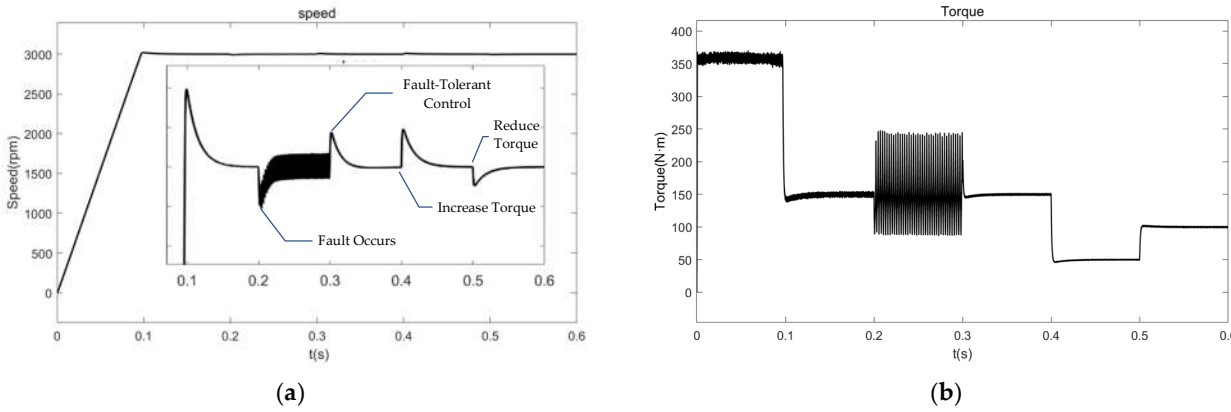

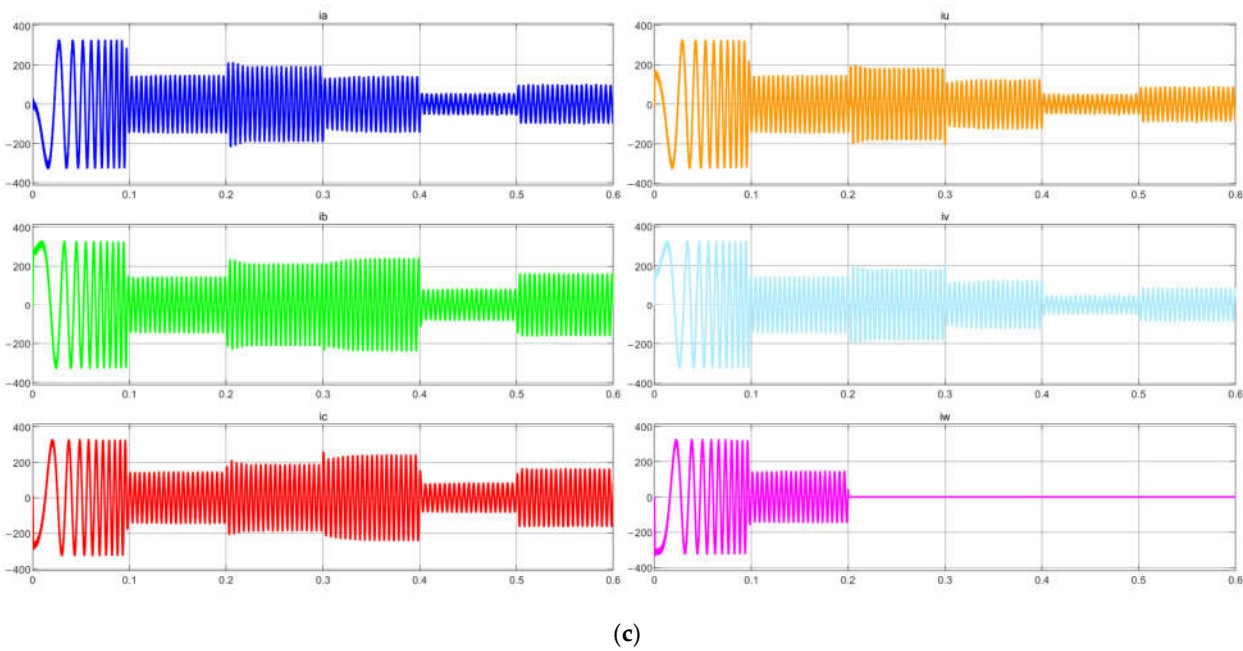

(**c**)

**Figure 8.** Simulation results of all periods under improved MRAS control. (**a**) Speed; (**b**) torque; (**c**) phase current.

## 5. Conclusions

This paper proposes a speed identification algorithm under fault-tolerant control. The simulation results show the effectiveness of this method. According to the simulation results of the ordinary motor, the faulty motor, and the motor under fault-tolerant control, this fault-tolerant strategy has suitable dynamic performance and steady-state accuracy. It ensures the stability of the whole system.

As for the improved MRAS method, the simulation results show that no matter whether an open-circuit fault occurs or not, it can accurately estimate the speed and angle of the motor, which expands the application scenarios of speed sensorless control. The novelty of this method is that it does not need to change the adjustable model in the MRAS and only needs to add some constraints to the output to realize the identification of the speed and angle under fault and fault-tolerant control, which is simpler and less computationally expensive. In the steer-by-wire system, this ensures that regardless of whether there is an open-circuit fault or a speed sensor fault, or even if both fail simultaneously, the steering operation can still be completed, which is of significant importance.

**Author Contributions:** Conceptualization, L.H. and G.D.; Methodology, C.F.; Validation, C.F. and L.H.; Formal analysis, C.F.; Investigation, C.F.; Resources, G.D. and M.Y.; Data curation, C.F.; Writing—original draft preparation, C.F. and L.H.; Writing—review and editing, C.F. and L.H.; Project administration, L.H., G.D. and M.Y.; Funding acquisition, M.Y. All authors have read and agreed to the published version of the manuscript.

**Funding:** This research was funded by Shanghai Automotive Industry Technology Development Foundation grant number 1811. And the APC was also funded by Shanghai Automotive Industry Technology Development Foundation.

**Acknowledgments:** The authors would like to thank Shanghai Automotive Industry Technology Development Foundation for supporting our team.

**Conflicts of Interest:** Yonggang Meng is the employee of Lianchuang Automotive Electronics. The paper reflects the views of the scientists, and not the company. The motor under test is provided by the company.

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
