# Peer review of "Sensorless Fault-Tolerant Control of Dual Three-Phase Permanent Magnet Synchronous Motor"

_wevj, doi:10.3390/wevj12040183_

Round 1
Reviewer 1 Report
The interesting paper studies the open-circuit fault-tolerant control strategy and speed identification algorithm under fault conditions. The research object is the dual three-phase permanent magnet synchronous motor. A sensorless operation under open-circuit fault is also provided by estimating speed using an improved model reference adaptive system (MRAS) speed estimator.
The research design is appropriate. The methods and results are adequately described. The results are clearly presented. The conclusions are supported by the results. The data are presented in tables, figures and equations correctly valued and interpreted in the paper. The references are properly indexed and recorded.
There are two "Figure 3.".
Figures 3. and 4. should be cross referred in the text. Use "is shown in Figure 3." instead of "is shown below:" in row 123 and the same with Figure 4 (second Figure 3.) in row 168.
Minor spell check required.
Reviewer 2 Report
Review of paper: Sensorless Fault-Tolerant Control of Dual Three-Phase Perma nent Magnet Synchronous Motor
Authors: Fan Cao, Haifeng Lu, Yonggang Meng, Dawei Gao
The paper present one type of fault-tolerant control strategies of control of multi-phase motor. The content of the article submitted for review is within the framework of the World Electric Vehicle Journal. However, the work requires revisions to enhance its merit.
List of critical comments:
- In my opinion, it is worthwhile for the authors to reconsider the linguistic correction of the article. This will eliminate minor linguistic and stylistic errors that were noticed in the text, and the content will be more accessible to the Readers (e.g. title of section 2, etc.). Please consider the professional service offered by MDPI.
- Please improve the abstract of the paper so that it fully specifies the content of the paper. Please do not use abbreviations in this section. There is room for that in the text of the article.
- Correct and complete the descriptions of the components of equations (1) and (2); failing to specify what the p parameter is? - lines 81-84.
- Equations are part of the text, just like the sentences that make it up. So put either a comma or a period at the end of each one. Please correct.
- Fig. 3 is included in the text 2 times in the text - correct. Please refer to specific figures by numbering them.
- Section 4 - Please describe in more detail the tools used in the simulation, the initial conditions and discuss the simulation results. The tests should concern a model situation when the drive is working under dynamically varying operating conditions (different variable load values), just like the engines of electrically driven cars are used.
- Please improve the closing statements in section Conclusion.
Reviewer 3 Report
This is a short paper that would better fit a Short Note type of manuscript.
The historical introduction can be interesting, however for a history journal.
Please improve the state of the art analysis to show the progress beyond the state of the art clearly. The lack of proper justification creates the wrong impression that the authors are unaware of the recent developments. Please use relevant recent references by OTHER authors, recent meaning from 2020 and 2021. This would provide the readers with a sense of continuity and help them place your paper in the context of what the journal has been publishing, very much strengthening your article's impact.
In the introduction, you need to connect the state of art to your paper goals. Please follow the literature review with a clear and concise state of the art analysis. This should clearly show the knowledge gaps identified and link them to your paper goals. Please reason both the novelty and the relevance of your paper goals.
In the conclusions, in addition to summarising the actions taken and results, please strengthen the explanation of their significance. It is recommended to use quantitative reasoning comparing with appropriate benchmarks, especially those stemming from previous work.
The originality of the paper needs to be further clarified. The present form does not have sufficient results to justify the novelty of a high-quality journal paper.
The results should be further elaborated to show how they could be used for real applications.
Please eliminate all multiple references. After that, please check the manuscript thoroughly and eliminate ALL the lumps in the manuscript. This should be done by characterising each reference individually. This can be done by mentioning 1 or 2 phrases per reference to show how it is different from the others and why it deserves mentioning.
Author Response
Please see the attachment。

Round 2
Reviewer 1 Report
After the first revision, they improved the paper in the right way.
They have added improvements to the abstract, introduction, and conclusions to make them more relevant and persuasive to the research. These changes did not influence the content and framework of the paper.
Accept the paper in present form.
Author Response
Thanks for your comments and suggestions sincerely.
I made some changes to the abstract and introduction of the paper according to the comments and suggestions of other reviewers. But these changes did not influence the content and framework of the paper.
The Word format file uses the “Track Changes” function, and the PDF format file is a clean manuscript.
We hope that our revised version will be satisfactory for publication in MDPI. Great thanks to you for the time and effort you expend on this paper.
Reviewer 2 Report
Review of paper:
Sensorless Fault-Tolerant Control of Dual Three-Phase Permanent Magnet Synchronous Motor
Authors:
Fan Cao , Haifeng Lu , Yonggang Meng and Dawei Gao
Comments:
Analyzing the responses of the authors of the paper, one can get the impression that most of the critical remarks have been properly taken into account and supplemented, and the text of the article has taken on a more valuable form for the Reader.
Thus, it can finally be concluded that the article in its revised form may be subject to a further publishing process.
Author Response
Thanks for your comments and suggestions sincerely.
We made some changes to the abstract and introduction of the paper according to the comments and suggestions of other reviewers.
We have made some supplements to the research methods and experimental results in the abstract to make it more substantial. We also made some changes to the introduction to make it more logical and organized. These changes did not influence the content and framework of the paper.
The Word format file uses the “Track Changes” function, and the PDF format file is a clean manuscript.
We hope that our revised version will be satisfactory for publication in MDPI. Great thanks to you for the time and effort you expend on this paper.
Reviewer 3 Report
The revision should be point-to-the-point answers.
Also both clean and mark manuscripts should be submitted.
So far the abstract should be improved:
Abstracts should contain 6 short items:
1) What is the problem being addressed?
2) What is the research question being asked?
3) What is the methodology being used to answer the stated research question?
4) What are the results obtained?
5) What is the meaning and importance of these results?
Please improve the state of the art analysis to show the progress beyond state of the art clearly. The lack of proper justification creates the wrong impression that the authors are unaware of the recent developments. Please use relevant recent references by OTHER authors, recent meaning from 2020 and 2021. This would provide the readers with a sense of continuity and help them place your paper in the context of what the journal has been publishing, very much strengthening your article's impact.
Author Response
Thanks for your comments and suggestions sincerely.
We made some changes to the abstract and introduction of the paper according to your comments and suggestions.
For the abstract part, we made some supplements to the research methods and experimental results to make it more substantial and balanced.
For the introduction part, we combed the development of fault-tolerant control over the past few decades and introduced the latest control methods in the past two years. Focusing on the speed sensorless control under fault-tolerant control, we introduced the rise and the latest research in the past two years. We hope this will be more logical and organized.
All of these changes don't influence the content and framework of the paper.
The Word format file uses the "Track Changes" function, and the PDF format file is a clean manuscript.
We hope that our revised version will be satisfactory for publication in MDPI. Great thanks to you for the time and effort you expend on this paper.